# Racism, homophobia, and the sexual health of young Black men who have sex with men in the United States: A systematic review

Sarah E. Janek[1]*, Lisvel A. Matos[1], Sandy Hatoum[2], Marta I. Mulawa[1,2], Leila Ledbetter[3], Michael V. Relf[1,2]

**1** School of Nursing, Duke University, Durham, NC, United States of America, **2** Global Health Institute, Duke University, Durham, NC, United States of America, **3** Medical Center Library, Duke University, Durham, NC, United States of America

* sarah.janek@duke.edu

**Data Availability Statement:** All relevant data are within the manuscript and its Supporting Information files.

## Abstract

Black gay, bisexual, and other men who have sex with men (BMSM) experience the highest rates of HIV acquisition annually out of any population in the United States, and young BMSM (YBMSM) are heavily impacted by this inequity as they enter adulthood. Despite a high annual HIV incidence, extant literature has found BMSM to engage in fewer sexual risk behaviors than White and Hispanic/Latino men who have sex with men, resulting in a gap between risk behaviors and the inequity of HIV infection. Structural factors, such as racism and homophobia, are thus being examined in order to understand this disconnect between behavior and HIV incidence. The purpose of this systematic review was to examine the discrimination experiences of YBMSM due to racism and homophobia in the United States and to evaluate the effect of these experiences on their sexual health. Four databases (MEDLINE, CINAHL Complete, APA PsycINFO, and Sociology Source Ultimate) were searched to examine the available qualitative, quantitative, and mixed method studies relevant to the research question. Out of 17 included studies, the majority were qualitative in design and were conducted in urban settings. Racism and homophobia affected YBMSM's sense of belonging, sexual identity, and sexual partnership choices. Often, masculinity would interact with these two constructs to impact how YBMSM engaged in sexual behavior, such as condomless sex, as well as their likelihood to seek sexual health care. Future research is needed to fully understand the relationships between discrimination and sexual health to develop effective structurally responsive interventions that will help decrease the inequities experienced by YBMSM.

## Introduction

In the United States, one in two Black gay, bisexual, and other men who have sex with men (BMSM) will acquire HIV within their lifetime compared to 1 in 11 White men who have sex with men [1]. Approximately three out of four HIV diagnoses for BMSM occur during the age

**Funding:** Research reported in this publication was supported by the National Institute of Mental Health of the National Institutes of Health under Award Number F31MH138075 (SEJ) and by the Duke University Center for AIDS Research CFAR), an NIH funded program (5P30 AI064518) (MVR). The content is solely the responsibility of the authors and does not necessarily represent the official views of the National Institutes of Health. The funders had no role in study design, data collection and analysis, decision to publish, or preparation of the manuscript.

**Competing interests:** The authors have declared that no competing interests exist.

range of 13 to 34 [2]. Given the profound HIV incidence and risk young BMSM (YBMSM) experience, sexual health and HIV prevention is important to improving the health of this population [2]. Extant literature has historically defined YBMSM beginning at 15 years of age, often citing the call for HIV screening beginning at 15 by the United States Preventive Services Task Force [3,4]. Thus, YBMSM will be defined as 15 to 34 years of age in this review, based on the prior and more contemporary epidemiologic evidence of HIV risk [1,2,4].

The World Health Organization (WHO) defines sexual health as, "a state of physical, emotional, mental, and social well-being in relation to sexuality; it is not merely the absence of disease dysfunction, or infirmity" [5]. Thus, sexual health care can be defined as care necessary to achieve this health and includes services such as HIV and sexually transmitted infection (STI) testing and medications for HIV pre-exposure prophylaxis (PrEP), such as emtricitabine-tenofovir disoproxil fumarate, a pill, or cabotegravir, an injectable form. Despite increased HIV rates for BMSM, extant literature has found BMSM to engage in fewer sexual risk behaviors, such as substance use during sex, and more preventive behaviors, such as condom use and increased lifetime HIV testing, than White and Hispanic/Latino men who have sex with men, resulting in a disconnect between their low levels of sexual risk behavior and high levels of HIV incidence [6–9]. Experiences of discrimination, particularly stemming from racism and homophobia, play a large role in this HIV inequity and other sexual health inequities, as YBMSM often face this same discrimination in the health care environment from providers and become hesitant towards seeking out future care [10–12].

Racism, although having many definitions, "refers to an organized system that categorizes population groups into 'races' and uses this ranking to preferentially allocate societal goods and resources to groups regarded as superior" ([13], p22). This type of discrimination, rooted in White supremacy, occurs at multiple levels, originating as systemic racism and trickling down to interpersonal racism. Systemic racism is often referred to interchangeably with structural or institutional racism, as the structure of society forms the institutions and policies that perpetuate discrimination and health inequities [13–15]. Within society, racism manifests on an interpersonal level, as both direct and indirect differential actions towards persons based on their race [15] and can become internalized through negative beliefs about a person's own race [16].

Homophobia is a form of sexual orientation discrimination based on prejudice, fear, and often hatred towards persons who engage in same-sex behavior [17]. Homophobia can often be referred to interchangeably with homonegativity, heterosexism, and sexual prejudice, and can occur across multiple levels similar to that of racism, including systemic and interpersonal, and sometimes taking root as internalized [17,18]. Experiences with feelings of shame and rejection associated with interpersonal homophobia can lead to internalized homophobia in YBMSM, presented as denial and negativity towards their sexual identity, as well as increased pressure on their outward display of masculinity to avoid further discrimination [6,19,20]. Regardless of the term chosen to represent this discrimination, there are detrimental health impacts on those who experience homophobia, including mental distress and avoidance of sexual health care such as HIV testing [6].

An intersectional framework guided this review to place the discrimination experiences and sexual health of YBMSM within the context of intersecting forms of oppression. Crenshaw's [21] theory of intersectionality underscores how systemic power structures in the United States, such as racism and homophobia, intertwine to perpetuate multiple levels of oppression against YBMSM, who face marginalization as both racial and sexual minoritized persons. While intersectionality acknowledges how the racial and sexual identities of YBMSM cannot be separated, and both identities influence their experiences and perceptions, this framework is especially crucial to understand the manifestation of discrimination. The intersection of systemic power structures presents in multiple levels of discrimination, such as

systemic and interpersonal, which collectively perpetuate health inequities for YBMSM [22]. Thus, to understand these inequities, we employed an intersectional framework to synthesize the intertwinement of racism, homophobia, and sexual health experienced by YBMSM.

Prior systematic reviews have focused on HIV interventions for BMSM across all ages [23], with some specifying structural barriers to HIV testing and prevention [3]. While these reviews have touched on sexual health considerations for BMSM, this review will be among the first to explicitly synthesize discrimination experiences (i.e., racism, homophobia) and sexual health for YBMSM. Through using an intersectional lens to analyze prior research, we will also fill a critical gap in the current literature that is necessary to understanding and mitigating health inequities. Thus, this review aims to understand how experiences of racism and homophobia impact the sexual health and sexual health care of YBMSM in the United States.

## Methods

### Design

We reported a systematic review of qualitative, quantitative, and mixed methods studies in accordance with the Preferred Reporting Items for Systematic Reviews (PRISMA) 2020 guidelines [24]. The PRISMA Checklist is presented in S1 Table. A registration protocol for this review was submitted to PROSPERO (CRD42022334014) prior to conducting the searches and screening. Addendums to the PROSPERO protocol were submitted and accepted as *sections*. For example, inclusion and exclusion criteria became more specified with title and abstract screening.

### Information sources

We searched MEDLINE via PubMed, CINAHL Complete via EBSCO Host, APA PsycINFO via EBSCO Host, and Sociology Source Ultimate via EBSCO Host to find research relevant to our research question.

### Search strategy

A search strategy was formed in collaboration with an expert research librarian (LL) at Duke University Medical Center Library. The search strategy included keywords of men who have sex with men; young adult; Black; discrimination, racism, and homophobia; and sexual health. These concepts and keywords were used to find MeSH terms, CINAHL subject headings, PsycINFO subject terms, and Sociology Ultimate subject terms that advanced each database's respective search. The searches were independently peer reviewed by another expert librarian using a modified PRESS Checklist [25]. The search was conducted by an expert librarian (LL) on May 20, 2022, re-run on March 13, 2023, and run for a final time on August 15, 2023 for the most current literature. A literature search table was completed for each database, forming a reproducible search strategy provided in S1 Appendix.

### Eligibility criteria

The question that guided this review is how racism and homophobia experiences affect the sexual health and sexual health care of YBMSM in the United States. The population of YBMSM was defined as Black or African American cisgender young adult men who have had oral and/or anal sex with a man at some point in their lifetime. Inclusion criteria based on sexual behavior allowed for inclusion of all sexual orientations that YBMSM may identify as, which allows for a more robust understanding of the discrimination and sexual health experiences of YBMSM. A broad criteria of Black or African American identity also allowed for the

inclusion of YBMSM who had additional racial or ethnic identities, such as those who identified as both Black and Hispanic/Latino, or other specific identities such as Afro-Caribbean American or Afro-Latino American. The young adult age range was defined as ages 15 through 34 years to account for BMSM before and while they enter an age range of higher HIV incidence [26]. This review included dates of data collection and publication since inception to include all potential extant literature discussing discrimination and sexual health of YBMSM. By including studies from inception, we aimed to capture all available literature discussing racism, homophobia, and sexual health since the available literature on this topic had been limited in preliminary searches. Additionally, inclusion since inception allows for understanding sexual health of YBMSM more broadly among evolving sociohistorical contexts and sexual health advancements.

Studies that focused on YBMSM as the population of interest were included. Included participants identified as gay/homosexual, bisexual, and other identifying sexual minority men. Studies also had to elaborate on participant sexual behavior, by either specifying inclusion of participants that had anal and/or oral sex with another man in their lifetime, or by providing descriptive or quantitative predictors of sexual behavior in the study. These criteria were created to include men who may identify as non-gay but still engage in sexual activity with other men. Thus, men who aligned with a non-straight sexual orientation, such as bisexual, but had not had oral and/or anal sex with men were excluded since they have different sexual health needs. While studies discussing men who have sex with men and women were included, studies that only involved men who only have sex with women were excluded. Due to this review's inclusion of HIV prevention as sexual health care, studies were included if they were comprised of a sample or a subsample solely of participants living without HIV. Studies were also eligible for inclusion if they included participants of unknown HIV serostatus, as they could engage in the same HIV prevention strategies as those known to be living without HIV. Studies were excluded if they included only participants living with HIV, or if the focus of the article was maintenance of HIV care and viral load suppression, as YBMSM living with HIV have specific sexual health care considerations. Studies with no subsample of YBMSM and rather, solely comprised of female participants, participants younger than 15 years of age, or participants not assigned male at birth were excluded. While it is important to work with participants of all genders, identities, and ages to advance health care and research, these populations may have different discrimination and sexual health experiences from cisgender YBMSM that would have deviated from this review's focus.

Studies were included if they explored the experiences of or relationships between discrimination, racism, and/or homophobia along with a sexual health outcome, with the population inclusion criteria stated above. Sexual health or sexual health care outcomes were considered actions that promoted or inhibited sexual health according to the WHO definition of sexual health [5], such as STI testing or condomless anal sex. Any sexual health outcome related to HIV, however, needed to be related to prevention. Studies solely focused on the relationship between stigma and sexual health outcomes were excluded, as stigma is a broad, overarching concept that would not be specific to understanding the experiences of racism and homophobia among YBMSM. Discrimination experiences of BMSM vary in other countries, especially those in which male-to-male sexual contact is illegal; thus, any studies published outside of the United States of America were excluded due to the focus on the discrimination experiences of YBMSM in the United States. Racism in the United States is also a unique construct due to its sociohistorical context and differs from other countries due to politics, cultures, and racial and ethnic diversity. There was no exclusion based on dates or types of publication to ensure adequate representation of existing literature, but publications had to be written in English to be included since the population of interest is within the United States.

Qualitative, quantitative, and mixed methods studies were included in this review. Editorials, abstracts, commentaries, and position pieces were excluded, as they would not include peer-reviewed findings that contribute to this review's aims. Studies focused on evaluating interventions through randomized controlled trials were also excluded, as these interventions could impact the outcome of sexual health and affect the answer to this review's research question. However, studies were eligible for inclusion if they solely reported baseline data before any randomization or intervention administration occurred. Instrument development and/or psychometric validity studies were also excluded due to a lack of relevance to the research question.

## Selection process

Identified studies (n = 2,379) were uploaded to Covidence [27], a review management software system that was used for screening, extraction, and quality assessments. Covidence [27] removed 690 duplicate studies, leaving 1,629 citations for two reviewers (SJ; LM) to screen by title and abstract, resulting in a moderate inter-rater reliability of κ = 0.60401. A third researcher (MR) was then used as an independent arbitrator for the final decision of inclusion or exclusion for any conflicts between researchers and to strengthen study reliability. After 165 studies were independently screened in full text (SJ; SH), with a moderate inter-rater reliability of κ = 0.52746, and conflicts were arbitrated (MR), 17 studies were included in this review. The screening process and resulting numbers of each stage of selection is outlined in the PRISMA 2020 flow diagram (Fig 1) [24].

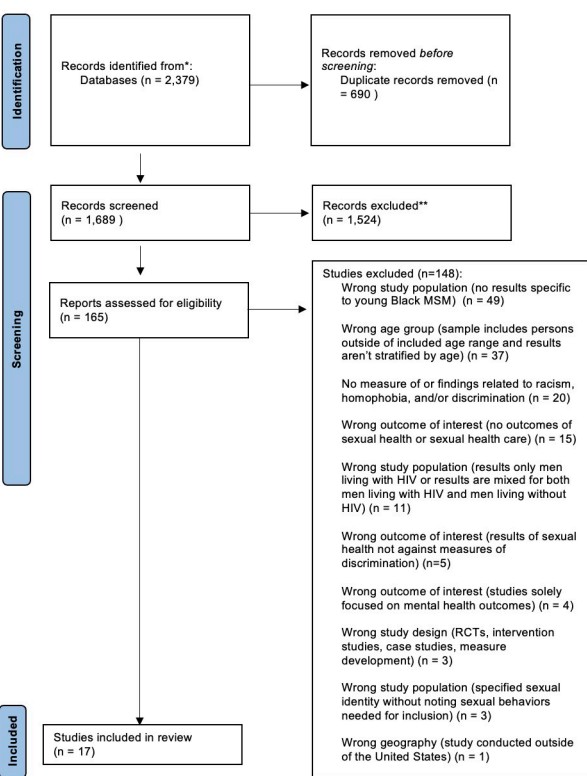

**Fig 1. PRISMA 2020 flow diagram for new systematic reviews which included searches of databases and registers only.**

## Data collection process

Each of the studies included after full-text review were read in detail, analyzed for key findings related to the research question, and assessed for quality. Extraction was performed by the primary reviewer (SJ) with validation performed by the other independent reviewer (LM) for every study. A data extraction form (ver. 2.0) was created in Covidence [27]. Each article's first author, year of publication, and study state (and city when available) were collected in the extraction process. We extracted the population of study and the subsequent age range, the sample size and any subsamples, the study design, and the study's stated research question(s) or objective(s). For qualitative studies, themes related to racism, homophobia, and sexual health were extracted, as well as relevant subthemes. Relevant exemplar quotes were provided for each theme, along with the corresponding page number, participant demographics when given, and the concept relevant to the research question, such as racism. For quantitative studies, we extracted the relevant variables and subsequent results in accordance with the research question. We used point estimates, effect sizes, confidence intervals, and *p*-values to report results and significance of statistical tests. We also noted any limitations discussed by the authors that may have impacted their results relevant to this review's research question.

## Study risk of bias assessment

To assess risk of bias and quality for each study, we utilized the QualSyst Quality Scoring of Qualitative Studies checklist and the QualSyst Quality Scoring of Quantitative Studies checklist [28]. If a study had mixed methods results, we would divide the risk of bias assessment based on type of data and use the corresponding checklist. Using a form created in Covidence [27], each reviewer independently assessed each study for risk of bias, and any disagreements in scoring were resolved through team discussion with the arbitrator. For the qualitative checklist, we used 10 items to assess how authors reported their design, methodology, results, credibility and reflexivity of their accounts [28]. For the quantitative checklist, we only included items for non-intervention studies, leaving 11 items for us to evaluate: authors' objectives, design, measures, sample size, analysis, and results [28]. For both checklists, a higher score indicated increased quality of studies. This assessment was not used to exclude articles from this review but rather to further contextualize the state of the extant literature by evaluating each study's rigor.

## Effect measures

For quantitative studies, we extracted the relevant coefficients for our main outcome and predictors, including point estimates, confidence intervals, and *p*-values examining their association, when provided by author. The effect sizes (e.g., Regression coefficients, Pearson's correlation coefficient, unadjusted and adjusted odds ratios and risk ratios), confidence intervals, and *p*-values were extracted for each association between the predictor with sexual health outcomes when included by author. For qualitative studies, we identified main outcomes through each study's key themes and exemplar quotes.

## Synthesis methods

Since racism and homophobia have not been thoroughly studied in this population and against measures of sexual health, there was anticipation of variation in reported variables and how relationships were reported. Thus, aggregated measures of effect were not feasible, and a descriptive synthesis of results was appropriate in understanding the state of the quantitative literature. JBI's [29] guidance was used for the narrative integration of both qualitative and

quantitative studies. Tables were used to report synthesis results for each study. Quotes reported both in text and tables followed the exact language as spoken by participants and reported by the authors of the included studies, even when participants used strong language.

# Results

## Summary of included studies

Out of the 17 included studies, 12 were qualitative and five were quantitative in design. No mixed methods studies met the inclusion criteria. Qualitative studies ranged in quality scores from 13 to 18 out of 20 using the QualSyst Qualitative checklist, and quantitative studies ranged in quality scores of 18 to 21 out of 22 after adapting the QualSyst Quantitative checklist for non-intervention studies [28]. The quality appraisal results of qualitative and quantitative studies are presented through robvis plots [30] in Figs 2 and 3, respectively. Qualitative studies received mostly full points across items, indicating high quality across design, analysis, and conclusions, but the majority of studies did not address reflexivity or credibility. Quantitative studies either received partial or full points on all items, indicating medium to high quality across multiple domains related to design, analysis, and results.

**Study sample, setting, design, and outcomes.** Sample size ranged from 25 to 52 participants for qualitative studies and from 120 to 1,210 YBMSM for quantitative studies. The majority of YBMSM participants across studies were between the ages of 18 and 29 years of age, regardless of study design [31–40]. As for geography, most studies (n = 15) were conducted in an urban setting [31,34–46]. Out of the qualitative studies, two were phenomenological [31,32], one was exploratory [33], one was grounded theory [45], and eight did not specify the type of qualitative methodology used [34–37,42–44,46]. All five quantitative studies were cross-sectional in design [38–41,47], but one [40] also included longitudinal, serial analyses. The majority of studies focused on sexual risk behaviors, such as condomless anal sex, as an outcome in relation to discrimination.

## Qualitative results

Across the qualitative studies, authors identified various themes pertaining to racism, homophobia, intersectional discrimination, and social support. All relevant author-identified themes (Table 1) were collapsed into broader categories for synthesis. Rich descriptions emerged about YBMSM engaging in sexual risk behaviors to compensate for the absence of social acceptance and support [31,36,42]. Another shared result pertained to YBMSM struggling with their sexual health and sexuality because of the difficulties of community acceptance due to their identities as both Black and men who have sex with men (MSM) [32,34,36,45]. Medical mistrust stemming from historical, structural oppression also influenced YBMSM's participation in sexual health care [32,33,43]. These key findings were revealed through synthesis of participant experiences, with exemplar quotes and rich descriptions guiding the reporting of results.

**Racism.** The experiences of YBMSM with racism stemmed from historical, systemic oppression resulting in medical mistrust and decreased sexual health resources [32,33,43]. YBMSM reflected on the systemic inequities of resources affecting their ability to think about and take PrEP, stating,

> I don't wanna say resources, because, like, everybody have the same, you know, seem like everybody got the same resources. I was thinkin' like resources, you know, 'cuz like you said, you know, the Black community, we gotta lot of stuff on our plate. And not to say, you know, Whites don't have a lot on their plate either but, you know, we're dealing with

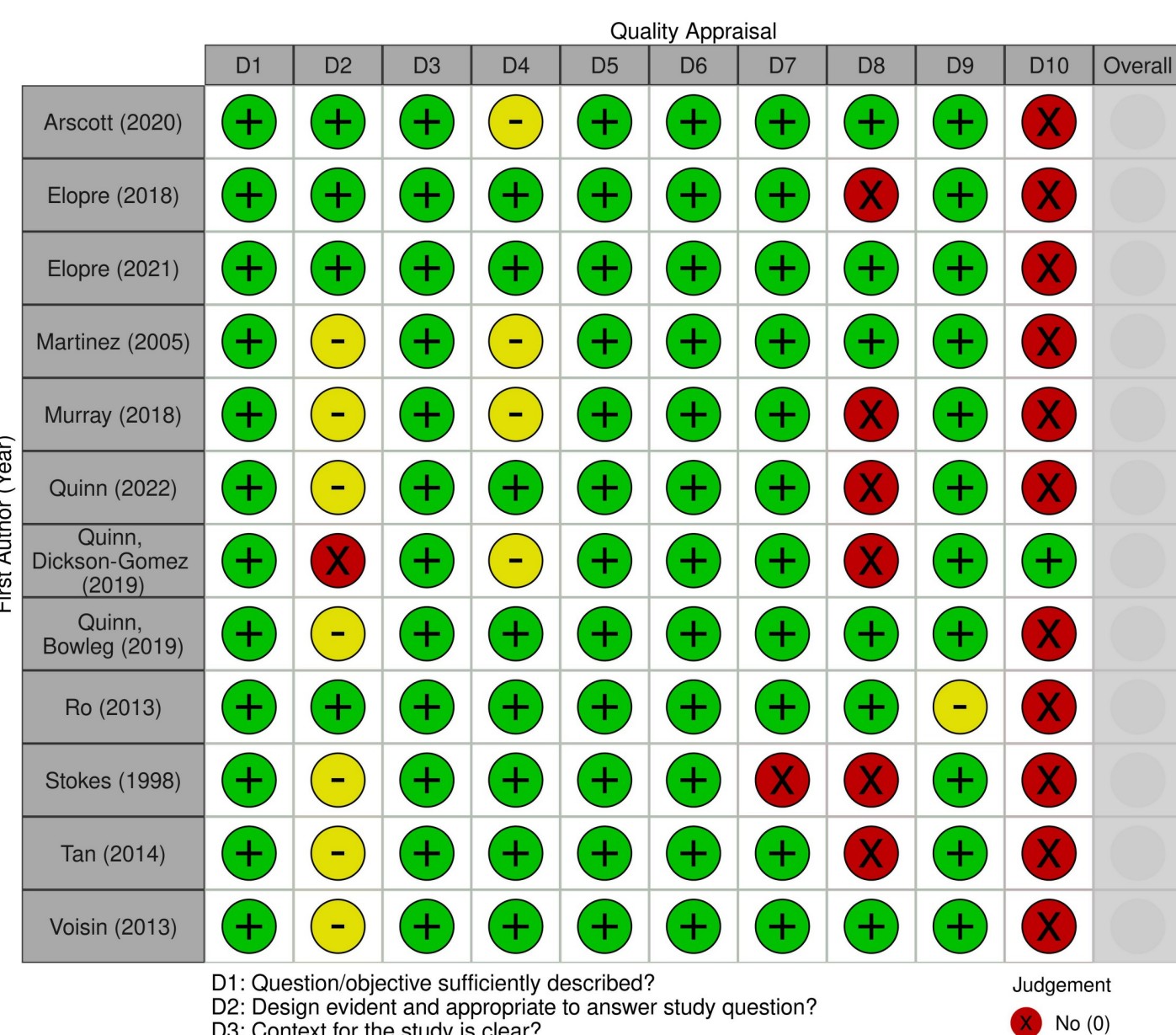

**Fig 2. Quality appraisal of qualitative studies.**

unemployment, finding jobs, you know, the hood. All the extra stuff, stuff. A lot of stuff that's on our plate, and so we not really carin' about PrEP, or whatever ([43], p90).

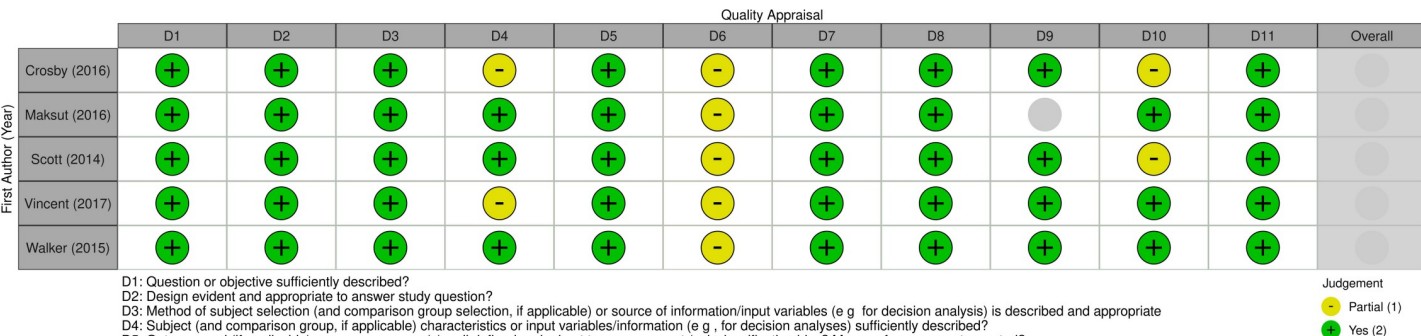

**Fig 3. Quality appraisal of quantitative studies.**

Systemic racism, presented in this quote as the differential allocation of resources, affected the ability of YBMSM to focus on their sexual health due to the other concerns, such as unemployment. Due to systemic and historical racism, YBMSM also relayed concerns of trust around PrEP. One participant referenced the mistreatment of Black Americans in health care and research historically, expressing,

> I'm so paranoid so this is to state the Tuskegee Syphilis Experiment and I always, in back of my mind, because of the history is healthy skepticism, that the shot may be to inject HIV into people and try to see what it does or it's not really to protect against HIV, maybe it's something else ([32], p515).

The participants in this study reflected on their fear of PrEP, and also mentioned how this information was passed down to them by older Black gay, bisexual, and other men who have sex with men, ultimately impacting how they would engage in HIV prevention [32].

YBMSM additionally felt they were expected to be at risk for or have HIV based on their racial identity. One YBMSM reflected on provided health information,

> I feel like, in general, people expect Black folks to be sickly in multiple capacities. Then when you get, I guess, years of information saying that like Black gay men have higher rates of STIs, and Black women are on the rise for having HIV transmission because of their positive partners, who are going to have sex with men, and not with their female partner ([33], p178).

Discordance between access to health care and expectations of sickness were presented throughout these studies, furthering distrust of sexual health care for YBMSM. Studies communicated how a long-standing history of power imbalances and misconduct in research and health care also affected the interactions between YBMSM and providers. YBMSM reflected on the concept of trust,

> Being a Black man is hard, period. I just don't trust other doctors or something that ain't my doctor. Like, search me, get me in a gown, none of that. It's a trust issue. And I just feel like, you know. I just feel like African Americans have it hard, period ([43], p88).

**Table 1. Results from qualitative studies of discrimination experiences and sexual health of YBMSM.**

| First Author & Year | Geography | Sample Size | Sample Age Range | Qualitative Design & Approach | Relevant Study Identified Themes |
|---|---|---|---|---|---|
| Arscott (2020) [33] | North Carolina, Maryland | N = 25 | 18-24 | Exploratory; directed content analysis | Sexual racism, discrimination, and stigma |
| Elopre (2021) [31] | Birmingham, Alabama | N = 25 | 18-29 | Phenomenology; inductive thematic saturation | Social support networks as important in validation of sexual identity |
| Elopre (2018) [32] | Alabama | N = 25 | 16-29 | Phenomenology; inductive and deductive thematic analysis | Stigma related to being Black, gay, and living in the South<br>Silence in the Black community about HIV prevention and sexual health<br>Medical distrust |
| Martinez (2005) [34] | Chicago, Illinois | N = 6 | 19-24 | Unspecified; coding and retrieving | Seeking healthcare |
| Murray (2018) [42] | New York City, New York | N = 108, n = 52 | 18-32 | Unspecified; inductive thematic analysis using constant comparison | Homophobia in the Black and Latino community<br>Fear of losing support from family and friends |
| Quinn (2022) [19] | Milwaukee, Wisconsin & Cleveland, Ohio | N = 46 | 18-37[b] | Unspecified; team-based intersectional thematic analysis | Social relationships |
| Quinn, Bowleg (2019) [43] | Milwaukee, Wisconsin | N = 44 | 16-25 | Unspecified; team-based, multi-stage thematic content analysis | Mistreatment within the health care system<br>Societal racism and inequity |
| Quinn, Dickson-Gomez (2019) [35] | Milwaukee, Wisconsin | N = 44 | 18-25 | Unspecified; team-based thematic content analysis | Young Black MSM[a] reduced to their sexuality by doctors<br>Structural disadvantage<br>Resistance to physicians' recommendations for PrEP |
| Ro (2013) [45] | Los Angeles County, California | N = N/A[c] | 18-60[b] | Grounded theory; analysis per Strauss and Corbin | Three dimensions of racism in the gay community: sexual stereotyping |
| Stokes (1998) [36] | Atlanta, Georgia, & Chicago, Illinois | N = 76 | 18-29 | Unspecified; no specified analysis type | Homophobia, self-esteem, and risk for HIV<br>Internalization of homophobia |
| Tan (2014) [46] | Los Angeles County, California | N = 35, n = 6 | 20-60[b] | Unspecified; analysis per Corbin and Strauss | Power exchanges that increase sexual risk-taking – sexuality for material resources |
| Voisin (2013) [37] | Chicago, Illinois | N = 10 | 18-24 | Unspecified; theoretical narrative analysis per Strauss & Corbin | Barriers to adopting HIV prevention behavior change |

[a]Men who have sex with men.

[b]Data only extracted from participants 15 to 34 years old.

[c]Not available from corresponding author.

The oppression experienced by YBMSM due to their Black identity heavily impacted their feelings about health care providers in general, as well as experiences in a health care setting. One participant reported feeling passed over in a health care setting while waiting to see a provider due to his Black identity stating,

I feel like that long waiting time, that feeling neglected at the hospital, that just all go with the passive aggressive racism that happens in certain states like Wisconsin. Whereas like in the South there's more direct racism, I feel like in Wisconsin it's more passive aggressive. Smile in your face, 'Hey, how you doin'?' But I'm gonna hold you down, type of racism. So it's like, while you in our hospital, I could serve you if you came in at 2 p.m., but I ain't serve you until like 6 p.m. because I got all these other people that came in that I'm gonna attend to first ([44], p1954).

This participant amplified the inequities experienced when YBMSM are finally able to seek care through recalling this act of interpersonal racism.

**Homophobia.** Homophobia primarily revolved around the sexual identities of YBMSM and their corresponding sexual health care [32,34]. YBMSM described their mistrust of health care settings and providers, specifically due to anticipation of homophobia, when one participant described how they felt about opening up to health care providers stating,

> ". . .it's all about how comfortable you're with somebody. When you first meet them, you're not gonna say well, I had sex with a man. Because first I want to figure out how they are going to judge me. The way society is now, homosexuality may be accepted. But, if you know that I've tapped into homosexuality, you would judge me a certain way, so I've got to be comfortable with you. . .It's kinda easy if you're comfortable with someone. So when you first meet someone, no [they won't disclose] – it's not until you break the ice with them that they'll actually tell you" ([34], p1109).

The disclosure of sexual identity or sexual behavior was heavily impacted by perceived and anticipated homophobia in health care settings, rooted in past experiences, and discomfort with disclosure affected how YBMSM sought care. YBMSM reflected on how providers treated them once they disclosed their sexual behaviors, with one YBMSM recalled his stereotyping experience with a primary health care provider,

> Right when you say you're gay it's like, STD check! Like right away it's like, I didn't come here for that. I gotta itch right here. Like, you know I got an itch on my head, do an STD check? What? No! ([35], p1955).

Both anticipated and experienced stereotypes about their gay identity or same-sex sexual behaviors served as a barrier for YBMSM to seek and receive care for any health concern, as providers would immediately assume they were there for a sexual health issue.

Concerns and experiences of homophobia translated from health care settings into the community. A YBMSM in Chicago reflected on how homophobia affected sexual health education and discussions,

> We gotta say we're privileged to be here. Cause you can be safe and be yourself. There's always fear that something's gonna happen to you, but we are in an area where we are well-informed about HIV. We have all of these centers to find out about HIV information. [The LGBT community] hit on it hard. A lot of these centers like, every Thursday we have a 'Let's Talk about Sex' so really they hit on HIV hard. Not everywhere, because everybody who is gay probably doesn't have resources like that. If you looked at one of the southern states, they are a little behind. They may not have that HIV information, it may not be available to them ([37], p114).

This participant discussed how sources of sexual health education and care changed based on geography due to less acceptance of lesbian, gay, bisexual, transgender, and queer (LGBTQ+) communities. This concern was mimicked in community settings in Alabama,

> I mean in the South, it's definitely I would say more difficult than anywhere in the LGBTQ community. Like being in the South, because I was raised in a real Christian environment, kind of overly religious when I think about it, but I just heard bisexuality is a cover up ([32], p514).

Both religion and geography played a role in how YBMSM were accepted and formed their own beliefs. These beliefs, in turn, affected YBMSM's comfort with their sexual identity and sexual health care.

**Intersectional discrimination.** The majority of studies [32,33,36,42–46] included results related to the intertwinement of racial and sexual identities, and consequently, racism and homophobia. These intersectional discrimination experiences presented in a variety of settings, including health care, universities, and nightclubs. Heavily impacting their sexual health, much of the discrimination YBMSM experienced in the health care setting involved negative stereotypes and pressure from providers based on both their racial and sexual minoritized identities [32,35,43]. One YBMSM recalled his provider assuming his health concern was related to having a sexually transmitted infection due to his identity as both a Black and gay man, "He just assumed because I was Black and I was gay, that this is what happened. . . I really, to be honest, I don't know I've been to the doctor since then" ([35], p1955). YBMSM felt immediately labeled as high risk due to being a Black man that has sex with another man, judged for their identities without evaluation of their own actions to keep themselves sexually healthy, resulting in their avoidance of health care. YBMSM also felt further frustration with this labelling due to the structural differences in allocation of sexual health resources based on race for gay men [32,35,43]. One participant reflected on their feelings toward the HIV inequity for YBMSM as,

> You know, white gay men are still white men. So they still, from their beginnings, are higher statistically like to be exposed to proper safe sex talks, safe sex training, proper health care talks, proper health insurance from their families, versus in the African American community you already have a disparity as far as healthcare and sex talk goes in general ([32], p514).

This lack of high-quality sexual health communication discouraged YBMSM from seeking care in the future, with YBMSM feeling hesitant due to fear of discrimination based on their identities and existing forms of intersecting oppression.

As YBMSM struggled with their relationships with sexual health care, they also felt lost in their search for acceptance and support, resulting in feelings of isolation in both the Black and LGBTQ+ communities [32,34,36,45]. Isolation from the LGBTQ+ community was recounted by a YBMSM who recalled racist interactions in a gay club,

> I've actually had someone in a club in West Hollywood actually call me the N word. . . When that happened, that really kind of like crystalized that there really is a very pervasive level of racism that exists within West Hollywood, and a lot of people don't want to talk about it. They don't want to acknowledge it ([45], p5).

Throughout this study [45], YBMSM recalled that although West Hollywood is a predominantly LGBTQ+ area, it was not welcoming to YBMSM due to their racial identity and YBMSM would suffer from racist, verbal attacks if they went out in that area, furthering isolation due to YBMSM also not having gay club options in the Black community. YBMSM in Chicago similarly felt a lack of belonging to LGBTQ+ communities, stating,

> I could be down there [in the predominantly gay area of Chicago] and feel like I don't fit in. Everyone there is white, feminine, and a couple of times I've been there, and it's a whole different world. It's a trip there – it's just not something I do ([34], p1109).

Although this participant experienced a more covert form of discrimination than the interpersonal racist attack in West Hollywood, YBMSM nonetheless experienced intersectional isolation and rejection from LGBTQ+ settings that typically serve as safe spaces for MSM.

Some YBMSM were cast out from Black communities as well, with rejection rooted in homophobia toward their sexual identities or behaviors [32,36,37,42,43]. YBMSM recounted their feelings of rejection due to their intersectional identities,

> I feel like a lot of us hide what it is that we're going through, or who we actually are, just because we can't be open about it. And that kind of hurts us even more because we don't go to the health departments and, I guess, get tested because we're afraid. And if we have it, who do we tell? Where do we find, I guess, that community? ([32], p514).

Feeling lost without a community presented major barrier to sexual health education and care for YBMSM, resulting in distress and care avoidance due to expectations of their identities across both LGBTQ+ and Black communities. Within Black communities, homophobia was associated with traditional forms of masculinity; YBMSM were rejected from their communities if they were not viewed as masculine or strong enough, and subsequently, thought to be gay [36,42,44]. One participant recalled growing up in this culture,

> Black men. . .are taught to be tough, not show emotions, to keep everything inside. And to show emotion [means' that you're a sissy, or you are a faggot. And growing up with that type of mind set, we tend to be homophobic, if you act a certain way, you're a sissy. If you say a certain thing or if you cry, you are a sissy. And we tend to be homophobic, taking on those values that if you show signs of weakness, that you're not a man ([36], p282).

These homophobic stereotypes of masculinity impacted the way YBMSM viewed their sexual identity as well as their own feelings, furthering distress and isolation.

Fear and isolation led some YBMSM to identify as "down-low" or "DL", rather than be gay-identified [34,36,42,43]. Comfort with their sexual identity and behaviors often affected how YBMSM engaged in risk behaviors, particularly relating to HIV/STI prevention [31,36,42]. A participant explained how YBMSM come to identify as down-low due to expectations of masculinity,

> the Black men, they tend to hide you know – most of them you know the 'DL' brothers you know what I'm saying because society doesn't accept two males being together, so it puts them at higher risks. And Black men are supposed to be strong, supposed to be the ruler of the house you know what I'm saying, and nobody – I don't think some Black men like to be called faggot or gay. It makes you feel belittle or whatever you know what I'm saying. So I think with society not accepting gay life or whatever, that put more Black men at risk of getting HIV because they gotta go to the park, they gotta go to places where people can't see them and go in the dark or whatever to do that's where you are more at risk too, we hide ([42], p183).

YBMSM reflected that Black gay-identified men or men who were "out" were more comfortable and more likely to practice safe sexual behaviors and engage in sexual health care due to feeling supported [34,36,42,43]. However, some gay-identified YBMSM participants relayed these same feelings of vulnerability affecting their preventive behaviors despite being "out" [44].

Gay-identified YBMSM struggled with feelings of rejection that would result in engaging in sexual risk behaviors regardless of their outward sexual identity. These participants still felt rejected by society due to their intersecting identities, with one YBMSM reflecting,

> I think that it's difficult because when you come out as gay you get, it's almost like your life gets thrown into this hole and you have to dig yourself out of it or else you will eventually die there. And a lot of us don't dig our way out of there. You know, it's like you get so many setbacks, so many fucking setbacks. . . . We [gay Black men] think we can survive without each other, but then we end up lonely and sad and then we get into either risky behavior or we become embittered to the world around us, and I don't want to be that ([44], pS408).

This reflection of YBMSM needing community and support due to structural, intersectional discrimination and rejection continued across studies. Feeling uncomfortable and rejected led some YBMSM to seek validation and social support through sex acts, disregarding prevention in exchange for feelings of acceptance [31,36,42,44]. A participant reflected on YBMSM and risk behavior, relaying, "I think that's why I think people have risky sex, really. Without them knowing it, they're trying to act out the validation that they don't get from people who they feel like they should get it from" ([31], p7). A participant in another study relayed this type of experience,

> I was so in love with this young man and so desperate to be loved and have a sense of belonging that I was willing to forgo the condom just to gain this man's love and acceptance. . .I was willing to contract this deadly disease just to prove my love to this young man or just to have him – for us to become one. [Interviewer asks, 'How long had you known him?'] Two weeks ([36], p288).

Despite only knowing this partner for a short period of time, this participant was willing to risk their sexual health for a chance at acceptance and belonging.

While YBMSM experienced societal and communal rejection due to their identities, they also found this rejection in seeking sexual partners [33,45]. YBMSM reflected on the difficulty of finding partners overall,

> I think that people are under the influence of like, 'Oh wow, that guy is gay and Black. That's a red flag. Just his existence is a red flag. I'm going to stay away from that because I don't want to become infected'. . . because of the intersection of my race and sexuality, which is perceived as also having a great increase for HIV ([33], p177).

YBMSM discussed sexual racism, either being labelled high-risk and unsafe by partners or hypersexualized due to being Black [33,45]. One YBMSM recounted the fetishization of Black men, hypersexualized as a "Mandingo fantasy" ([45], p8), with another participant mirroring these same feelings,

> I think too there's sometimes, you know, when it's the Mandingo factor I think some men get, initially they're frightened of what they're attracted to and, you know, it can come out as, that fear can come out as, you know, in racist ways. You know, Black people are going to steal my wallet, you know, whatever - it might not be spoken that way but there's, you know, you can kind of observe behavior. And then it might turn around to, you know, actually I want it or if you don't give it to them then, then the racist stuff can come out afterwards, you know ([45], p9).

These participants found partners to initially be interested in them, but then experience racist interactions after engaging in sexual behaviors with them. Another participant discussed this contradiction in sex work,

Well, it's only a sexual thing. It's not like we're going to the movies or I could go to their places of employment and try to get a job. In the real world it is a whole other ballgame, what people do and how people discriminate, they may do these things [like pay you for sex], but these things are all behind closed doors and only they know about it, but in the real world they would never sit next to you on a bus. They would never even come to your aid. Yet [they] would still go to a black prostitute's house and patronize her business or his business or an escort's house ([46], p6).

Intersectional discrimination affected sexual partnerships, not only in how non-YBMSM viewed YBMSM, but also how they stereotyped other YBMSM as high-risk sexual partners.

YBMSM struggled with the internalization of discrimination due to the amount of discrimination they experienced both societally and interpersonally from sexual partners. A YBMSM participant discussing his partner selection perpetuated the same racial stereotypes he experienced himself, joined with an additional stereotype about socioeconomic status, ". . .like if I'm having sexual relations with someone who is African American and gay and from a low-income family, then I feel like that also makes my chances of catching it [HIV] go higher" ([33], p177). This participant stereotyped potential partners not only based off of their sexual and racial identities, but also based on sociodemographic factors such as socioeconomic status, adding the intersecting power structure of classism to label partners as higher risk. Another participant reflected how religion plays a role in the internalization of homophobia for YBMSM,

If [churchgoing men] are sexual with someone of the same sex, they're probably gonna feel guilty afterwards. And they're probably gonna have low self-esteem and not have much confidence in themselves because they are things that have been preached against ([36], p285).

YBMSM found themselves feeling guilty for their sexual identity and behaviors due to the community and society around them. The internalization of sexual racism and homophobia not only affected how YBMSM selected partners, but also fed into internal conflict about their identities and their navigation through intersecting power structures. Another participant reflected his dilemma as,

. . .it's really, really strange way of thinking, the way that I do, but it makes me feel more inclined to vet Black people when I engage in sexual practices. And I am just like, is this anti-Black? Is this, like, racist that I am doing this? Or is this warranted because in the way in which anti-blackness has created a scenario in which it's more likely for like Black people to have HIV? So, like, am I am doing good health practicing, or am I participating in anti-Black racism right now. I can't tell right now. And it's really frustrating. And so, [I'm] always thinking about that ([33], p181).

YBMSM across studies recalled this inner conflict due to fear of potentially perpetuating the same structures and stereotypes of anti-Black racism that society and potential sexual partners placed onto them. This internalization also caused mental health distress and decreased self-esteem, which would ultimately affect how YBMSM participated in sexual health prevention.

Self-esteem, impacted by discrimination, heavily influenced how YBMSM engaged in sexual behavior and sexual health care. YBMSM often felt acquiring HIV was inevitable based on how they were treated by society and sexual partners,

> [We need] to be loved. If all your life you're told that you're going to hell or you're going to be damned or whatever, you begin to take that into consideration. You begin to feel, 'Damn, well, I'm going to hell because I'm this way.' Every time we look up, someone's saying something negative about us. If you don't love yourself. Nobody can love you. So what do you care about getting any [information] if you are no good anyway ([37], p116).

Due to the oppression and discrimination YBMSM had already experienced across their lifespan, such as the lack of resources and upheld stereotypes, YBMSM often felt discouraged and demotivated towards protecting their sexual health.

## Quantitative results

All five quantitative studies [38–41,47] focused on sexual risk behaviors as the main outcome of interest, with most variables concerning condom use and anal sex. All five studies analyzed a construct related to homophobia. While some studies adhered to the same construct of homophobia [39] and internalized homophobia [38,41], others referred to a different variable under this umbrella construct, such as heterosexism [40] and sexual identity centrality or regard [47]. Walker et al. [47] additionally focused on racism through the constructs of racial centrality, racial private regard, and racial public regard. Despite these different construct names, however, all studies were focused on race and/or sexuality-related discrimination, with how YBMSM viewed their own identities and how they felt others viewed their identities as an aspect of their sexual health. Two studies considered social support as a predictor or moderator [39,40]. A description of the predictor variables, study outcomes, effect sizes, confidence intervals, and p-values for statistical tests conducted in each study can be found in Table 2.

**Internalized homophobia.** Two of the five quantitative studies included internalized homophobia as a predictor of sexual health in YBMSM [38,41]. Crosby et al. [41] found greater level of internalized homophobia resulted in a lower likelihood of being tested for HIV in the past 12 months as well as lower likelihood to engage in condomless oral sex. However, Maksut et al. [38] did not find a significant relationship between internalized homophobia and condomless anal sex among their YBMSM subsample. Thus, studies found varying results for how internalized homophobia impacted the sexual health of YBMSM, with results either being significant for both promotive and inhibitive sexual health behaviors or nonsignificant findings regarding the relationship between internalized homophobia and sexual health behaviors.

**Intersectional relationships.** Two studies [39,47] analyzed the intersectional nature of YBMSM identities by jointly studying both race and sexuality variables. Walker [47] found that centrality of YBMSM's sexual identity was negatively associated with how they felt the public viewed their race. Sexual identity public regard and racial identity public regard were positively correlated, displaying congruence between how participants felt the public viewed both their sexual and racial identities. The total number of both anal sex acts and unprotected (i.e., condomless) anal sex acts were negatively associated with racial identity centrality and public regard. This association indicated that participants engaged in fewer total sex acts and fewer unprotected sex acts when they had greater positive feelings about one's race by the public and its importance to their identity. Similarly, Scott et al. [39] found that more frequent experiences of racism and homophobia were associated with delayed HIV testing. In their hierarchical logistic regression model, structural discrimination (racism and homophobia

**Table 2. Results from quantitative studies of discrimination experiences and sexual health of YBMSM.**

| First Author & Year | Geography | Sample Size | YBMSM[a] Sample Age Range | Study Design | Analysis | Predictor | Outcome | Regression & Correlation Coefficients | OR/ AOR/ RR [95% CI] | *p* value |
|---|---|---|---|---|---|---|---|---|---|---|
| Crosby (2016) [41] | Midsize Southern U.S. city | N = 595; n = 470 | 15-29 | Baseline data from a RCT[b] | Bivariate | Internalized homophobia | HIV testing in the prior 12 months | NR[d] | NR | 0.07[†] |
| | | | | | | | Willing to take PrEP | NR | NR | 0.60 |
| | | | | | Multiple logistic regression[c] | Internalized homophobia | Unprotected anal sex as a bottom | NR | AOR: 1.69 [1.13-2.53] | 0.01[‡] |
| | | | | | | | Sex with women | NR | AOR: 3.43 [2.13-5.52] | <0.001[‡] |
| | | | | | | | Used condoms with most recent sex partner | NR | AOR: 1.37 [0.92-2.02] | 0.12[‡] |
| | | | | | | | Tested for HIV in the past 12 months (adjusted) | NR | AOR: 1.55 [1.01-2.30] | 0.04[‡] |
| | | | | | | | Discussed AIDS prevention with sex partners | NR | AOR: 1.54 [1.09-2.15] | 0.01[‡] |
| | | | | | | | Discussed mutual sex histories with sex partners | NR | AOR: 1.33 [0.90-1.97] | 0.15[‡] |
| | | | | | | | Not identify as MSM to health care providers | NR | AOR: 1.89 [1.15-3.12] | 0.01[‡] |
| | | | | | | | Engaged in condomless oral sex | NR | AOR: 0.64 [0.41-0.99] | 0.049[‡] |
| | | | | | | | Tested RPR+ | NR | AOR: 0.66 [0.40-1.09] | 0.1[‡] |
| Maksut (2016) [38] | Atlanta, Georgia | N = 450 BMSM, n = 230 YBMSM | 18-29 | Correlational | Univariate regression | Internalized homophobia | Condomless anal sex | NR | RR: 1.06 [0.96-1.16] | NS[e] |

*(Continued)*

**Table 2.** (Continued)

| First Author & Year | Geography | Sample Size | YBMSM[a] Sample Age Range | Study Design | Analysis | Predictor | Outcome | Regression & Correlation Coefficients | OR/ AOR/ RR [95% CI] | p value |
|---|---|---|---|---|---|---|---|---|---|---|
| Scott (2014) [39] | Dallas & Houston, Texas | N = 813 | 18-29 | Cross-sectional | Bivariate | Racism | Delayed HIV testing | NR | OR: 1.20 [1.02-1.41] | 0.026* |
| | | | | | | Homophobia | | NR | OR: 1.49 [1.02-2.17] | 0.038* |
| | | | | | | Social support from other BMSM | | NR | OR: 0.80 [0.67-0.95] | 0.01* |
| | | | | | Hierarchal multivariate logistic regression | Combined structural discrimination (racism + homophobia) | Delayed HIV testing (>6 months prior) | NR | NR | 0.019* |
| | | | | | | Combined structural discrimination + social support from other BMSM | | NR | NR | 0.037* structural; 0.012* social support |
| | | | | | | Combined structural discrimination + social support + individual sociodemographics (age, socioeconomic distress, unprotected anal intercourse with a non-concordant partner) | | NR | NR | 0.206 structural; 0.047* social support; <0.001* individual |
| Vincent (2017) [40] | Dallas & Houston, Texas | N = 1,210 | 18-29 | Cross-sectional, longitudinal serial | Multiple linear regression | Social support x internalized heterosexism x enacted heterosexism | Condom self-efficacy | b = .001 | [<.001, .001] | 0.01 |
| | | | | | | Social support x internalized heterosexism | | b = .003 | [-.001, .006] | .101 |
| | | | | | | Social support x externalized heterosexism | | b = .001 | [-.001, .003] | .308 |
| | | | | | | Internalized heterosexism x enacted heterosexism | | b = -.005 | [-.007, -.002] | <.001*** |
| | | | | | | Internalized heterosexism | | b = -.133 | [-.189, -.077] | <.001*** |
| | | | | | | Enacted heterosexism | | b = -.227 | [.284, -.170] | <.001*** |
| | | | | | | Social support | | b = .058 | [.004, .112] | .035 |

(*Continued*)

**Table 2.** (Continued)

| First Author & Year | Geography | Sample Size | YBMSM[a] Sample Age Range | Study Design | Analysis | Predictor | Outcome | Regression & Correlation Coefficients | OR/ AOR/ RR [95% CI] | p value |
|---|---|---|---|---|---|---|---|---|---|---|
| Walker (2015) [47] | Nationwide | N = 120 | 18-29 | Cross-sectional | Bivariate | Racial identity centrality | Total number of anal sex acts | $r = -.20$ | NR | <0.05* |
| | | | | | | | Sexual identity centrality | $r = .35$ | NR | <0.001** |
| | | | | | | Racial public regard | Total number of anal sex acts | $r = -.22$ | NR | <0.05* |
| | | | | | | | Sexual identity centrality | $r = -.23$ | NR | <0.05* |
| | | | | | | Racial identity public regard | Sexual identity public regard | $r = .29$ | NR | <0.05* |
| | | | | | Hierarchal linear regression | M1: Outness | Total anal sex acts | $R^2 = .002$ | NR | .631 |
| | | | | | | | Total unprotected anal sex acts | $R^2 = .009$ | NR | .375 |
| | | | | | | M2: Outness + sexuality centrality + sexuality private regard + sexuality public regard | Total anal sex acts | $R^2 = .041$ $\Delta R^2 = .039$ | NR | .242 |
| | | | | | | | Total unprotected anal sex acts | $R^2 = .043$ $\Delta R^2 = .034$ | NR | .415 |
| | | | | | | M3: Outness + sexuality centrality + sexuality private regard + sexuality public regard + **Racial centrality** + racial private regard + **racial public regard**[f] | Total anal sex acts | $R^2 = .144$ $\Delta R^2 = .103$ | NR | 0.008* |
| | | | | | | | Unprotected anal sex acts | $R^2 = .137$ $\Delta R^2 = .094$ | NR | 0.042* |

Note: indicates author identified significance

*** at < 0.001

** at 0.01

* at 0.05

† at 0.1

‡ at 0.2.

[a] Young Black men who have sex with men.

[b] Randomized controlled trial.

[c] Adjusted for age and HIV status.

[d] Not reported.

[e] Author did not provide significance.

[f] Significant variables within the model are bolded.

combined) was significantly associated with delayed HIV testing. Thus, in quantitative studies measuring both sexual and racial identities, there were significant correlations between these constructs, as well as significant associations between the constructs and sexual health behaviors.

**Social support.** The study by Scott et al. [39] also found when receiving social support from Black gay or bisexual male friends, participants were less likely to report delayed HIV testing even when accounting for structural discrimination (racism and homophobia combined). The effect of social support on delayed HIV testing remained significant even when controlling for individual factors (e.g. age, socioeconomic distress, unprotected anal sex with a non-concordant partner). In the final model, however, the relationship between individual measures of experiences of racism and homophobia were not statistically significantly associated with delayed HIV testing, though the combined measure of structural discrimination remained significantly associated with delayed HIV testing.

Another study, conducted by Vincent et al. [40], found YBMSM who reported high levels of internalized and enacted heterosexism also reported low levels of condom self-efficacy, indicating a negative association between heterosexism and condom self-efficacy. Both internalized and enacted heterosexism were also found to modify the relationships between social support and condom self-efficacy. Specifically, when examining adjusted three-way interactions between social support, condom self-efficacy, and both internalized heterosexism and enacted heterosexism, the significance of the relationship between social support and condom self-efficacy varied based on the level of both internalized and enacted heterosexism, after adjusting for sociodemographic factors. When social support was at an average level, condom self-efficacy remained average in the midst of high internalized and enacted sexism. However, condom self-efficacy scores increased to above average when social support was average in the midst of high internalized heterosexism and low enacted heterosexism, as well as when both internalized and enacted heterosexism were low [40]. As a result of these two studies [39,40], social support was found to significantly positively affect HIV testing and condom use behaviors in YBMSM, even in the midst of experiencing discrimination.

## Discussion

This review was among the first to take an intersectional lens towards synthesizing the extant literature surrounding the impact of discrimination on the sexual health of YBMSM. By conducting a review of both quantitative and qualitative research, we uncovered key findings about the state of the science surrounding YBMSM's sexual health in the United States, allowing insight into the sexual health inequities experienced by this population. While limited quantitative studies conducted analysis of race- and sexuality-related variables together, qualitative studies highlighted the intersectional challenges YBMSM faced as being both Black in the LGBTQ+ community and LGBTQ+ in the Black community [32,34,36,45]. Homophobia that YBMSM experienced in the Black community was heavily impacted by religion [32,36] and expectations of masculinity [36,42,44]. YBMSM also experienced discrimination due to intersecting demographics such as socioeconomic status [33] and geography [32,37]. This struggle to belong to a community ultimately affected how YBMSM felt potential sexual partners viewed them and how YBMSM felt seeking sexual health care.

Both quantitative [39,47] and qualitative data [34,36,42,43] described the impact of discrimination on sexual identity development and subsequently, on the sexual health of YBMSM. Sexual identity and racial identity were heavily intertwined as Walker et al.'s [47] findings of the significant relationship between sexual identity and racial identity bolstered the qualitative results that focused on YBMSM's intersection of identities and the corresponding challenges [32,33,36,42–46]. Qualitative studies depicted YBMSM's search for a community that accepts and validates YBMSM identities [32,36,37], with social support being quantitatively presented as protective against sexual risk behaviors [39,40]. Experienced discrimination often manifested in stereotypes from providers [34,35,43] and sexual partners [33,45,46], resulting in

identity issues [36,37,42], internalized discrimination [33,36], and avoidance of sexual health care [35]. Interestingly, there was variation in how internalized homophobia influenced risk behaviors, with it decreasing likelihood of health care behaviors such as recent HIV testing while also decreasing likelihood of sexual risk behaviors [38,41]. Quantitative results of decreased sexual risk behaviors [41] contrasted with the qualitative results that suggested YBMSM with greater internalized homophobia engaged in more unprotected sex acts [31,34,36,42,43]. This variation across study types and variables advocates for additional research to better understand the heterogeneity of intersectional discrimination experiences across YBMSM and subsequently, how this variation ties into understanding the sexual health of YBMSM.

This review builds on the synthesis of previous reviews focused on MSM and their HIV care across the United States. While other reviews have synthesized the interventions focused on HIV prevention and treatment across MSM [48] and BMSM [23] or systemic barriers to care for BMSM [3], this review offers a holistic viewpoint of sexual health impact resulting from discrimination experiences. Although previous reviews have uncovered how stigma affects HIV prevention and treatment across MSM [49], YBMSM experience discrimination in unique and heterogenous ways that affect their sexual health care and thus, require research focused specifically on their age group. Our findings also narrowed the age range of previous reviews [3,23], focused on BMSM during an age range critical for sexual health prevention and narrowing the gap of research and interventions needed. Thus, this review contributes a synthesis that offers important considerations for future research and interventions specific to YBMSM.

Insight into the intersecting sources of oppression, such as racism and homophobia, that drive the discrimination experiences of YBMSM and the subsequent impact on their sexual identity formation, risk behaviors, and sexual health care engagement lays a foundation for future, structurally responsive interventions. For example, an intervention formed to counter LGBTQ+ and Black community rejection and provide health education for YBMSM found promising results, including YBMSM wanting to become certified HIV peer counselors to promote testing in their community [50]. This intervention bolstered the importance of social support found in this review and the motivation by YBMSM to be sexually healthy and engage in care, emphasizing the importance of interventions to promote this well-being. Structural interventions that specifically address intersecting sources of power and oppression, such as those that target unemployment or housing instability, have additionally proved to be positive paths for advancing health equity across the United States [51]. YBMSM disproportionately struggle with housing discrimination [50] among the already disparate access to resources due to systemic oppression, affecting their ability to seek sexual health care and engage in HIV prevention [52]. Both policy reform and intervention development can aid in mitigating these structures of power, beginning with the accessibility of basic resources, such as stable housing, and building a pathway for YBMSM to receive non-discriminatory, affordable, and comprehensive sexual health care [53].

YBMSM frequently experience discrimination once accessing health care services [32,35,43,44], which points to the potential for both multi-level interventions (i.e., combining structural policy reform with institutional interventions in health care environments) and health care setting-specific interventions. These structural and institutional interventions could utilize strategies such as engaging health care providers in education and patient simulations to reduce implicit bias and discrimination [54]. Regardless of the approach, interventions across multiple levels are needed to combat the intersectional discrimination that YBMSM face in everyday life, in a variety of settings, to ultimately improve their sexual health.

## Limitations

Despite a rigorous and systematic protocol, this review has several limitations. This review's rigor may have been impacted due to some qualitative studies not specifying their design, which would have allowed for a richer understanding of the state of the science and methodology. Quantitative studies varied in their approaches to determining the significance of their statistical tests, determining their results to be significant or nonsignificant using different criteria, such as setting alpha at 0.2 instead of the traditional 0.05. The varying levels may have affected the synthesis of significant results across studies, as results synthesized in this review were deemed significant based on each study's original criteria and results. Thus, systematic aggregation of results across studies for this field of study proves difficult and reinforces the importance of using standardized levels of significance. Since literature, particularly qualitative studies, displayed the heterogeneity in YBMSM experiences with discrimination, future quantitative studies are warranted to explore this heterogeneity using person-centered approaches that may identify subgroups of YBMSM with differing patterns of intersectional discrimination levels and sources.

Additionally, although this review sought to understand sexual health more broadly, many of the quantitative studies included focused on sexual risk behaviors, such as condomless anal sex. Thus, future quantitative research focusing on discrimination should focus on or continue to include additional, more promotive outcomes, such as PrEP use or STI testing frequency. This future avenue for research can also build on the sexual health advancements that have occurred since the included studies were conducted, such as injectable PrEP. Gaining more quantitative research with a sexual health promotion focus would aid in addressing this gap and further the understanding of the connection between sexual health care behaviors and HIV inequities within the context of health care innovation.

Another noted limitation was the inconsistency of inclusion and exclusion criteria of YBMSM across studies. Many studies had to be excluded for using inclusion criteria that defined YBMSM as a man having a gay or bisexual identity or participating in sexual behavior with another man without any outcomes related to sexual behavior. Since a person who identifies as gay or bisexual does not necessarily engage in sexual behaviors, we excluded those studies due to the possibility of the sample never having participated in sexual behavior that may affect their discrimination and sexual health experiences. However, future research may benefit from a more expansive definition of YBMSM, allowing for inclusion based on sexual orientation alone.

Studies used various terms to describe and measure both the discrimination experiences and sexual health of YBMSM. While this review had intended to look for constructs of racism and homophobia, additional constructs emerged (i.e. biphobia, heterosexism) that became synonymous with this sexuality-related discrimination. In the reviewed quantitative studies, we found significant heterogeneity in the operationalization of these constructs, thus affecting the ability to aggregate results across studies. Additionally, while variables of and results related to internalized homophobia were expected, internalized racism emerged as a key concept across multiple types of studies, proving the internalization of racial discrimination to be instrumental in understanding YBMSM's sexual health as well. We uncovered this difficulty in reviewing qualitative studies as well due to the nuances of themes related to discrimination and sexual health causing ambiguity surrounding inclusion. We believe that this limitation is the main reason that our inter-rater reliability was only moderate; while we strived for maximum reviewer agreement, studies investigating and explicitly referencing discrimination as key predictors or themes are limited and difficult to determine.

Lastly, the majority of the synthesized studies were conducted in urban environments, leaving a gap for research focused on discrimination and sexual health in rural settings to be explored. Extant literature has displayed different considerations for MSM seeking PrEP and disclosing their sexual identity to providers in rural areas due to differences in LGBTQ+ discrimination experiences and access to care [55–57]. As some studies found differences in sexual health and discrimination in the South [32,37], it is important for research to account for geography to understand how these experiences differ across regions and could impact sexual health care.

## Conclusion

This review is among the first to explore how the racism and homophobia experiences of YBMSM impact their sexual health and sexual health care. By using an intersectional lens to article synthesis, this review places the intertwinement of minoritized identities at the center of the sexual health of YBMSM. Although this review sought to focus on the racism and homophobia experiences of YBMSM, factors of religion, geography, and socioeconomic status also emerged as key determinants of discrimination experiences and as a result, sexual health. This review highlights the importance of community and peer support to YBMSM well-being and sexual health, and how social relationships may mitigate the effects of discrimination, providing a positive path for future research.

Future research is needed not only to standardize the tools used to measure these types of discrimination but is also needed for both qualitative and quantitative exploration of the effects of intersectional discrimination on YBMSM's sexual health. Additional studies are needed to fully understand the relationships between discrimination and sexual health as both racism and homophobia were rarely the main constructs of interest in studies, and sexual risk behaviors were often the outcomes of interest in quantitative studies. Intersectionality offers a lens to begin dismantling power structures, as well as understand discrimination experiences stemming from these structures that ultimately drive sexual health inequities among YBMSM . Future research focused specifically on the systemic roots of discrimination and sexual health promotion can help to promote health equity through further identifying gaps for interventions and policies most needed and relevant to the health and well-being of YBMSM.

Ultimately, extant literature found that racial and sexual identity discrimination, defined using multiple constructs, affected YBMSM's sexual health in various ways, ranging from feelings about their sexual identity to sexual behaviors. Racism, homophobia, and often the intersection of the two, affected how YBMSM viewed their sexual identity which often corresponded to their sexual risk behaviors and sexual health care seeking, as well as their engagement in sexual health education and prevention efforts.

## Supporting information

**S1 Table. PRISMA 2020 checklist.**
(PDF)

**S1 Appendix. Search strategies.**
(PDF)

## Acknowledgments

We would like to gratefully acknowledge Dr. Rosa Gonzalez-Guarda's and Dr. Sara LeGrand's guidance in the formation of this review's protocol. We additionally would like to thank Dr.

Allison Stafford, Dr. Julee Waldrop, and Ms. Elena Turner for their assistance in editing this manuscript.

## Author Contributions

**Conceptualization:** Sarah E. Janek, Leila Ledbetter, Michael V. Relf.

**Data curation:** Sarah E. Janek, Lisvel A. Matos, Sandy Hatoum, Leila Ledbetter, Michael V. Relf.

**Formal analysis:** Sarah E. Janek, Lisvel A. Matos, Marta I. Mulawa, Michael V. Relf.

**Funding acquisition:** Michael V. Relf.

**Investigation:** Sarah E. Janek, Lisvel A. Matos, Sandy Hatoum, Leila Ledbetter, Michael V. Relf.

**Methodology:** Sarah E. Janek, Leila Ledbetter, Michael V. Relf.

**Project administration:** Sarah E. Janek.

**Resources:** Sarah E. Janek, Marta I. Mulawa, Leila Ledbetter.

**Software:** Sarah E. Janek, Leila Ledbetter.

**Supervision:** Sarah E. Janek, Marta I. Mulawa, Michael V. Relf.

**Validation:** Sarah E. Janek, Lisvel A. Matos, Michael V. Relf.

**Visualization:** Sarah E. Janek, Lisvel A. Matos, Marta I. Mulawa.

**Writing – original draft:** Sarah E. Janek.

**Writing – review & editing:** Sarah E. Janek, Lisvel A. Matos, Sandy Hatoum, Marta I. Mulawa, Leila Ledbetter, Michael V. Relf.

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
