## [Decision Letter · Decision Letter 0]

17 Jul 2024

PONE-D-24-03457Racism, homophobia, and the sexual health of young Black men who have sex with men in the United States: A systematic reviewPLOS ONE

Dear Dr. Janek,

Thank you for submitting your manuscript to PLOS ONE. After careful consideration, we feel that it has merit but does not fully meet PLOS ONE’s publication criteria as it currently stands. Therefore, we invite you to submit a revised version of the manuscript that addresses the points raised during the review process. I would like to apologise for the delay in the peer-review of this manuscript. It has been challenging to find suitable peer reviewers with availability. You can find the detailed reviews below. Overall, the study is scientifically sound, but some issues need addressing. The use of "risky sexual behavior" is problematic and should be replaced. The inclusion of all manuscripts since inception requires justification, particularly concerning language changes pre- and post-2012 with the advent of PrEP. The introduction should incorporate a broader discussion of PrEP, including both pill-based and injectable forms, and emphasise the role of power in intersectionality. In the results section, certain phrases need refinement for clarity and accuracy. The discussion should deepen its focus on intersectionality, emphasising the role of power in interpreting data and policy implications. 

We look forward to receiving your revised manuscript.

Kind regards,

Daniel Demant, PhD, MPH, GradCertHEd, BAppSocSc

Academic Editor

PLOS ONE

Reviewers' comments:

Reviewer's Responses to Questions

**Comments to the Author**

1. Is the manuscript technically sound, and do the data support the conclusions?

Reviewer #1: Yes

Reviewer #2: Yes

2. Has the statistical analysis been performed appropriately and rigorously? 

Reviewer #1: N/A

Reviewer #2: N/A

3. Have the authors made all data underlying the findings in their manuscript fully available?

Reviewer #1: Yes

Reviewer #2: Yes

4. Is the manuscript presented in an intelligible fashion and written in standard English?

Reviewer #1: Yes

Reviewer #2: Yes

5. Review Comments to the Author

Reviewer #1: This is an interesting review of the relationship between racism and homophobia on sexual health of YBMSM. Overall, this paper is well written and makes a valuable contribution. I have 2 overarching concerns/comments and some more editorial suggestions.

In your introduction and throughout the paper you cite the absence of any evidence of greater sexual risk to explain disparities in HIV infection as justification for looking at racism and homophobia. However, many of the papers reviewed used sexual risk behaviors (e.g., condomless anal sex) as the outcome. While homophobia and racism may be linked to HIV disparities, it should not be expected to be mediated through sexual risk behaviors that we have already determined are not greater than those of non-Black MSM. At best, a relationship between experiences of discrimination or internalized stigma and sexual risk might explain variation in HIV prevalence among YBMSM. Health care seeking behaviors such as length of time between HIV tests, for example, make more sense as a potential explanation for the paradox you present in the introduction. I am not suggesting that you not review papers that explore the link between homophobia, racism and sexual risk, just to point out the limitations of these studies in understanding disparities.

I appreciated the detailed description of the qualitative papers. However, the way the results were written made it sometimes hard to know if the interpretations following quotes were yours or the original authors.

In the methods, you name "phenomenological, exploratory, grounded theory" as methods when they are maybe better described as analytic perspectives/frameworks.

On line 361, some-thing, doesn't need the hyphen

443 "Comfortability" should be replaced by comfort

542 "This internalization also caused mental health distress and self-esteem" It seems like there is a verb missing before self-esteemed (lowered?).

655 Consider replacing juxtaposed with the more direct "contrasted with". Juxtapose is sometimes used just to specify a side-by-side comparison without suggesting a sharp contrast.

Reviewer #2: The authors present a systematic review of literature to understand the relationship between racism, and homophobia, and sexual risk among young Black men who have sex with men (YBMSM). The authors have conducted the work scientifically, but there are some concerns with the choices made, and considerations for the authors in detail below.

Overall:

I think there is a need to specify similar but differently understood language. “Risky sexual behavior” tends to be used in a stigmatizing and personal (individual) way given the imprecision of the language, whereas sexual risk behavior is an objective indicator which can be measured. I ask the authors to consider this throughout the manuscript.

The choice to include all manuscripts since inception seems odd – and requires the authors to explain changes in language that do not currently appear in the manuscript. Specifically, before 2012 (the approval of PrEP), many researchers did use the term “unprotected” when sex was condomless, with a partner living with HIV, or a partner of unknown status; however, we have ceased using this term because now YBMSM can have condomless sex that is not "unprotected" given viral suppression, HIV PrEP, and Doxy PEP use. I believe that this either needs to be prominently in the introduction or in the discussion with limitations to the data and attention should be paid to this change throughout the manuscript.

Introduction:

Pp3, lines 58-59: I wondered why the authors named Truvada rather than explaining pill-based and injectable versions of PrEP. According to the methods, injectable PrEP was available before the start of this project (2021).

Does line 67-69 (pp 3) need a page number?

Pp 4-5, lines 87-94: I am really pleased to see intersectionality here, but this use of intersectionality is less focused on power (social power, social inequality, and social justice) as tenets of intersectionality. Including the importance of power when using this lens may strengthen the author’s discussion and conclusion. It may even help to point out the policy/intervention needs of the populations.

Methods:

The methods section was well-written.

I would ask the authors: any considerations as inclusion of Afro-Latinos?

Pp 9, line 193: why do the authors think the kappa score was so low? What does this mean for how the study was conducted? If there are implications to the kappa being so low, that should be explained in the limitations.

Pp11, line 241: Please spell out JBI, please.

Results:

Pp 15, lines 304-306: Do the authors mean older YBMSM? This phrase makes the sentence awkward, and I am not sure it is the implied meaning of the original author.

Pp 21, line 421: I suggest "[Some] YBMSM were cast out…" as there are sources that show that this is not an issue solely in Black communities

Discussion:

The discussion follows the data found in the review; however, I believe an increased focus on intersectionality not just as a lens for the population selection, but also in the interpretation of the data. Parts of the discussion (i.e., pp 36, lines 679-685) could serve as a space where the importance of power within the use of intersectional theory could be infused. I remind the authors that the purpose of intersectionality is critical praxis – it’s about the doing of something and not merely the ability to describe colocations of identity.

6. PLOS authors have the option to publish the peer review history of their article (what does this mean?). If published, this will include your full peer review and any attached files.

Reviewer #1: No

Reviewer #2: No

---

## [Author Response · Author response to Decision Letter 0]

28 Oct 2024

Dear Reviewers,

We thank you for your thoughtful review and contributions towards our manuscript’s successful revision. We have revised our manuscript to address all editorial and reviewer comments, as outlined below:

Editorial Comments

1: “The use of ‘risky sexual behavior’ is problematic and should be replaced”. 

RESPONSE: We appreciate the reviewers’ dedication to non-stigmatizing language throughout our manuscript. We have removed this phrase in our text with the exception of when this language is used in direct quotes from participants in the original studies. For example, we have changed “risky sexual behaviors” to “sexual risk behaviors” in line 302, and changed “would result in risky sexual behaviors” to “would result in engaging in sexual risk behaviors” in lines 500-501, and changed “more risky, unprotected sex acts” to “more unprotected sex acts” in line 723.

2: “The inclusion of all manuscripts since inception requires justification, particularly concerning language changes pre- and post-2012 with the advent of PrEP”. 

RESPONSE: We chose to include articles since inception due to our focus on sexual health and HIV prevention broadly, rather than within the context of PrEP. While the arrival and increasing access to of PrEP was critical for this population, we felt that it was important to highlight all of the available literature around racism, homophobia, and sexual health over the years since a review has not been previously been conducted on this topic. Please see our response to Reviewer 2’s second recommendation for more detail. 

3: “The introduction should incorporate a broader discussion of PrEP, including both pill-based and injectable forms, and emphasize the role of power in intersectionality”.

RESPONSE: We have provided additional detail around PrEP in the introduction and discussion. We have also incorporated greater focus on power in intersectionality in accordance with Reviewer 2’s comments (see responses to Comments 2, 3, and 1 for Reviewer 2 below). 

Reviewer 1:

1: “In your introduction and throughout the paper you cite the absence of any evidence of greater sexual risk to explain disparities in HIV infection as justification for looking at racism and homophobia. However, many of the papers reviewed used sexual risk behaviors (e.g., condomless anal sex) as the outcome. While homophobia and racism may be linked to HIV disparities, it should not be expected to be mediated through sexual risk behaviors that we have already determined are not greater than those of non-Black MSM. At best, a relationship between experiences of discrimination or internalized stigma and sexual risk might explain variation in HIV prevalence among YBMSM. Health care seeking behaviors such as length of time between HIV tests, for example, make more sense as a potential explanation for the paradox you present in the introduction. I am not suggesting that you not review papers that explore the link between homophobia, racism and sexual risk, just to point out the limitations of these studies in understanding disparities”.

RESPONSE: We appreciate this thoughtful comment and are grateful for the opportunity to clarify the rationale for our review question and focus of our manuscript. We did not intend to imply that we expected the relationship between homophobia/racism and HIV disparities to be mediated through sexual risk behaviors. Our intention was to study the relationship between homophobia, racism, and sexual health more broadly (i.e., not focusing only on sexual risk behaviors). We strove to include the consideration of sexual health care seeking behaviors as contributors to this inequity, introducing this concept on page 4, and discussing these behaviors from the included studies throughout the manuscript (see pages 19, 21, 32-33, and 35). 

While our inclusion criteria were broad, our review predominantly included studies focused on risk behaviors as an outcome, especially quantitative studies, due to the limitations of currently published literature. Unfortunately, there were few quantitative studies that moved beyond these behaviors as a focus, and we have added this discussion to our limitations in lines 793 to 802. 

2: “I appreciated the detailed description of the qualitative papers. However, the way the results were written made it sometimes hard to know if the interpretations following quotes were yours or the original authors”.

RESPONSE: We appreciate this comment and have reviewed and edited our qualitative synthesis accordingly. In the results section, interpretations made by the original authors were summarized, though our synthesis provided additional connections across studies. We have made minor edits to the results section to clarify any ambiguity in the source of interpretations throughout the manuscript.

3: “In the methods, you name "phenomenological, exploratory, grounded theory" as methods when they are maybe better described as analytic perspectives/frameworks”.

RESPONSE: We chose to describe these methodologies as such due to the categorization among qualitative research experts. We used the description of qualitative research methodologies offered by Sage Research Methods to inform our decision: https://doi.org/10.4135/9781473920163.

4: “On line 361, some-thing, doesn't need the hyphen”.

RESPONSE: Thank you to Reviewer 1 for noticing this typo, and we have removed the hyphen in line 395. 

5: “443 "Comfortability" should be replaced by comfort”

RESPONSE: We appreciate the language recommendation provided and have changed the word to “comfort” in line 483. 

6: “542 ‘This internalization also caused mental health distress and self-esteem’. It seems like there is a verb missing before self-esteemed (lowered?)”.

RESPONSE: We thank the reviewer for this observation and have included the change to “decreased self-esteem” in line 595-596.

7: “655 Consider replacing juxtaposed with the more direct "contrasted with". Juxtapose is sometimes used just to specify a side-by-side comparison without suggesting a sharp contrast”.

RESPONSE: This recommendation very thoughtful; we have replaced the phrase in line 722.

Reviewer 2:

1: “I think there is a need to specify similar but differently understood language. ‘Risky sexual behavior’ tends to be used in a stigmatizing and personal (individual) way given the imprecision of the language, whereas sexual risk behavior is an objective indicator which can be measured. I ask the authors to consider this throughout the manuscript”.

RESPONSE: Thank you for this thoughtful language consideration. We have switched the language to sexual risk behavior, with the exception of included quotes of study participants using the phrase. As we described in response to Editorial Comment 1 above, we have changed “risky sexual behaviors” to “sexual risk behaviors” in line 302, and changed “would result in risky sexual behaviors” to “would result in engaging in sexual risk behaviors” in lines 500-501, and changed “more risky, unprotected sex acts” to “more unprotected sex acts” in line 723.

2: “The choice to include all manuscripts since inception seems odd – and requires the authors to explain changes in language that do not currently appear in the manuscript. Specifically, before 2012 (the approval of PrEP), many researchers did use the term ‘unprotected’ when sex was condomless, with a partner living with HIV, or a partner of unknown status; however, we have ceased using this term because now YBMSM can have condomless sex that is not ‘unprotected’ given viral suppression, HIV PrEP, and Doxy PEP use. I believe that this either needs to be prominently in the introduction or in the discussion with limitations to the data and attention should be paid to this change throughout the manuscript”.

RESPONSE: We appreciate this consideration being pointed out. While year is an important consideration, this review’s focus was general sexual health and HIV prevention outcomes. Our review sought to synthesize all aspects of sexual health, not restricted to HIV, and YBMSM are still at increased risk of other sexually transmitted infections through condomless sex regardless of partner viral suppression, PrEP, or PEP use. Thus, we did not limit the years to that since PrEP was approved since our focus was not solely on PrEP. We found it important to include studies from inception to understand how discrimination has been used as a predictor or construct of interest over time and among sexual health advancements. Since sexual health studies have often not incorporated the sociohistorical context experienced by YBMSM specifically, we felt this wide date range was of greater priority to the field of research rather than limiting the scope to one aspect of sexual health. 

We also anticipated that many studies would be more recent because of the lack of studies specifically looking at discrimination based on our prior, preliminary search. The three studies that discussed “unprotected” anal sex (Crosby et al., 2016; Scott et al., 2014; Walker et al., 2015) all were published after 2012, and thus, studied after the widespread approval of PrEP. We kept the authors’ original language of “unprotected” when describing the study results to ensure accurate reporting of results. However, we have clarified when authors defined “unprotected” as “condomless”, notably in line 660 for Walker et al. (2015). We have also added additional clarity to our choice of inclusion dates in lines 157 through 161.

3: “Pp3, lines 58-59: I wondered why the authors named Truvada rather than explaining pill-based and injectable versions of PrEP. According to the methods, injectable PrEP was available before the start of this project (2021)”.

RESPONSE: We chose to include emtricitabine-tenofovir disoproxil fumarate as an example of PrEP. We have differentiated the pill form of PrEP, as well as added an example of the injectable PrEP in line 59. 

We would also like to note that injectable PrEP was approved in December of 2021 and our first search was conducted five months later in May of 2022. While we re-ran the search multiple times in 2023, the only study that met our inclusion criteria was that of Quinn et al. (2022), which included data from 2019. Thus, none of our studies collected data since the FDA approval of injectable PrEP and we did not feel that an in-depth discussion of this modality was relevant to the synthesis of our included studies. However, we have discussed this as a future avenue for research and future reviews to consider in lines 793-802. 

4: “Does line 67-69 (pp 3) need a page number?”

RESPONSE: Thank you to Reviewer 2 for catching this detail; we have added a page number for this quote in line 70. 

5: “Pp 4-5, lines 87-94: I am really pleased to see intersectionality here, but this use of intersectionality is less focused on power (social power, social inequality, and social justice) as tenets of intersectionality. Including the importance of power when using this lens may strengthen the author’s discussion and conclusion. It may even help to point out the policy/intervention needs of the populations”.

RESPONSE: We appreciate this reviewer’s attention to the roots of intersectionality and wholeheartedly agree that the intersection of oppressive powers should be highlighted. We have added additional language surrounding this in the introduction, from lines 89 to 104, and the discussion throughout lines 740 to 777. We have also incorporated more language about this structural oppression throughout the qualitative results, further shifting the language from intersecting identities to more of a structural oppression focus. 

6: “I would ask the authors: any considerations as inclusion of Afro-Latinos?”

RESPONSE: We thank the reviewer for this thoughtful question. We included any studies that included YBMSM that identified as Afro-Latino, as we considered this identity to be included under the umbrella criteria of Black or African American. Additionally, only one of the included studies (Quinn, Bowleg et al., 2019) had participants that identified as both Black and Hispanic/Latino. To add clarification, we have specified our inclusion in lines 151-154 as follows: 

“By including studies from inception, we aimed to capture all available literature discussing racism, homophobia, and sexual health since the available literature on this topic had been limited in preliminary searches. Additionally, inclusion since inception allows for understanding sexual health of YBMSM more broadly among evolving sociohistorical contexts and sexual health advancements.”

7: “Pp 9, line 193: why do the authors think the kappa score was so low? What does this mean for how the study was conducted? If there are implications to the kappa being so low, that should be explained in the limitations”.

RESPONSE: Although our kappa score for both title and abstract and full text screening was moderate in accordance with established guidelines of 0.41-0.60 (Landis and Koch, 1977; McHugh, 2012), we believe this is due to the nature of the studies reviewed and synthesized. As discussed throughout our review, studies explicitly calling out racism and homophobia were difficult to discern, and often studies would allude to these constructs in their discussion rather than as a predictor or theme in their study. While perfect agreement is always the goal, qualitative data in particular have unique challenges for assessing inclusion and exclusion criteria. Based on the complexity of these data and having the majority of our included studies be qualitative, we did not feel a high inter-rater reliability was likely. 

Additionally, we sought guidance from other reviews of this nature, cited throughout our manuscript, but the other reviews did not discuss inter-rater reliability among authors or provide a kappa score. Despite this, we opted to provide this score for transparency and have added a discussion about this limitation (see lines 819 to 824). We have also adjusted our language in lines 213 through 217 to add clarity that arbitration with a third, independent researcher occurred after the inter-rater reliability was already established between the two initial, independent reviewers. Thus, we used arbitration as a way to strengthen reliability in the inclusion and exclusion of studies.

8: “Pp11, line 241: Please spell out JBI, please”.

RESPONSE: We appreciate this attention to acronym definition. However, based on the JBI Manual for Evidence Synthesis (see https://doi.org/10.46658/JBIMES-24-01) and other guidelines, JBI is no longer spelled out as The Joanna Briggs Institute and is rather just cited by the acronym. Thus, we have left this acronym as is.

9: “Pp 15, lines 304-306: Do the authors mean older YBMSM? This phrase makes the sentence awkward, and I am not sure it is the implied meaning of the original author”.

RESPONSE: The study cited in these lines (Elopre et al., 2018) highlighted the challenges of misinformation and advice from older Black gay, bisexual, and other men who have sex with men in their communities. We have changed the phrase from “older YBMSM” to “older Black gay, bisexual, and other men who have sex with men” to provide additional clarity in lines 335-336.

10: “Pp 21, line 421: I suggest "[Some] YBMSM were cast out…" as there are sources that show that this is not an issue solely in Black communities”.

RESPONSE: Thank you for this suggestion; we have added “Some” in line 460.

11: The discussion follows the data found in the review; however, I believe an increased focus on intersectionality not just as a lens for the population selection, but also in the interpretation of the data. Parts of the discussion (i.e., pp 36, lines 679-685) could serve as a space where the importance of power within the use of intersectional theory could be infused. I remind the authors that the purpose of intersectionality is critical praxis – it’s about the doing of something and not merely the ability to describe colocations of identity.

RESPONSE: We are grateful for this recommendation and agree that the language around intersectionality should be more social justice oriented, rather than mainly focus on identities. Please see R5 Response for more detail

---

## [Decision Letter · Decision Letter 1]

13 Dec 2024

Racism, homophobia, and the sexual health of young Black men who have sex with men in the United States: A systematic review

PONE-D-24-03457R1

Dear Dr. Janek,

We’re pleased to inform you that your manuscript has been judged scientifically suitable for publication and will be formally accepted for publication once it meets all outstanding technical requirements.

Kind regards,

Daniel Demant, PhD, MPH, GradCertHEd, BAppSocSc

Academic Editor

PLOS ONE

Additional Editor Comments (optional):

The authors have addressed all comments sufficiently. The remaining two comments are related to minor grammar issues that are not sufficient to warrant an additional round of revisions as these can be amended during the production stage.

Reviewers' comments:

Reviewer's Responses to Questions

**Comments to the Author**

1. If the authors have adequately addressed your comments raised in a previous round of review and you feel that this manuscript is now acceptable for publication, you may indicate that here to bypass the “Comments to the Author” section, enter your conflict of interest statement in the “Confidential to Editor” section, and submit your "Accept" recommendation.

Reviewer #1: (No Response)

2. Is the manuscript technically sound, and do the data support the conclusions?

Reviewer #1: Yes

3. Has the statistical analysis been performed appropriately and rigorously? 

Reviewer #1: Yes

4. Have the authors made all data underlying the findings in their manuscript fully available?

Reviewer #1: Yes

5. Is the manuscript presented in an intelligible fashion and written in standard English?

Reviewer #1: Yes

6. Review Comments to the Author

Reviewer #1: A few nitpicky things. Since you changed risky sexual behavior to sexual risk behavior, "less" should now be replaced with "fewer".

Apologies for not pointing this out sooner if this was part of the last version, but I found the explanation for excluding papers that focused on stigma "broadly" very confusing. Stigma is something that happens as a result of belonging to a discredited group, so if not race or sexual identity, what were these general stigma questions measuring? HIV stigma? Just naming the other kinds would be sufficient.

line 725 "that may identified" I think should be "who may identify."

7. PLOS authors have the option to publish the peer review history of their article (what does this mean?). If published, this will include your full peer review and any attached files.

Reviewer #1: No

---

## [Editor Report · Acceptance letter]

6 Jan 2025

PONE-D-24-03457R1 

PLOS ONE

Dear Dr. Janek, 

I'm pleased to inform you that your manuscript has been deemed suitable for publication in PLOS ONE. Congratulations! Your manuscript is now being handed over to our production team.

Kind regards, 

on behalf of

Dr. Daniel Demant 

Academic Editor

PLOS ONE